# Does Sodium Intake Induce Systemic Inflammatory Response? A Systematic Review and Meta-Analysis of Randomized Studies in Humans

**DOI:** 10.3390/nu13082632

**Published:** 2021-07-30

**Authors:** Eirini D. Basdeki, Anastasios Kollias, Panagiota Mitrou, Christiana Tsirimiagkou, Marios K. Georgakis, Antonios Chatzigeorgiou, Antonios Argyris, Kalliopi Karatzi, Yannis Manios, Petros P. Sfikakis, Athanase D. Protogerou

**Affiliations:** 1Cardiovascular Prevention & Research Unit, Clinic & Laboratory of Pathophysiology, Department of Medicine, National and Kapodistrian University of Athens, 11527 Athens, Greece; eirinibasdeki@gmail.com (E.D.B.); tsirimiagou.ch@gmail.com (C.T.); and1dr@gmail.com (A.A.); 2Department of Nutrition and Dietetics, School of Health Science and Education, Harokopio University of Athens, 17671 Kallithea, Greece; manios@hua.gr; 3Hypertension Center STRIDE-7, Third Department of Medicine, School of Medicine, National and Kapodistrian University of Athens, Sotiria Hospital, 11527 Athens, Greece; taskollias@gmail.com; 4Hellenic Republic Ministry of Health, 10433 Athens, Greece; mitroup@moh.gov.gr; 5Institute for Stroke and Dementia Research, University Hospital, Ludwig Maximilians University, 81377 Munich, Germany; mgeorgakis91@gmail.com; 6Department of Physiology, Medical School, National & Kapodistrian University of Athens, 11527 Athens, Greece; achatzig@med.uoa.gr; 7Department of Food Science and Human Nutrition, Agricultural University of Athens, Iera Odos 75, 11855 Athens, Greece; pkaratzi@aua.gr; 8First Department of Propaedeutic and Internal Medicine, Athens University Medical School, Laiko Hospital, 11527 Athens, Greece; psfikakis@med.uoa.gr

**Keywords:** sodium, sodium intake, inflammation, systemic inflammation, sodium induced inflammation, CRP, TNF-a, IL-6

## Abstract

Experimental studies suggest that sodium induced inflammation might be another missing link leading to atherosclerosis. To test the hypothesis that high daily sodium intake induces systemic inflammatory response in humans, we performed a systematic review according to PRISMA guidelines of randomized controlled trials (RCTs) that examined the effect of high versus low sodium dose (HSD vs. LSD), as defined per study, on plasma circulating inflammatory biomarkers. Eight RCTs that examined CRP, TNF-a and IL-6 were found. Meta-analysis testing the change of each biomarker in HSD versus LSD was possible for CRP (*n* = 5 studies), TNF-a (*n* = 4 studies) and IL-6 (*n* = 4 studies). The pooled difference (95% confidence intervals) per biomarker was for: CRP values of 0.1(−0.3, 0.4) mg/L; TNF-a −0.7(−5.0, 3.6) pg/mL; IL-6 −1.1(−3.3 to 1.1) pg/mL. Importantly, there was inconsistency between RCTs regarding major population characteristics and the applied methodology, including a very wide range of LSD (460 to 6740 mg/day) and HSD (2800 to 7452 mg/day). Although our results suggest that the different levels of daily sodium intake are not associated with significant changes in the level of systemic inflammation in humans, this outcome may result from methodological issues. Based on these identified methodological issues we propose that future RCTs should focus on young healthy participants to avoid confounding effects of comorbidities, should have three instead of two arms (very low, “normal” and high) of daily sodium intake with more than 100 participants per arm, whereas an intervention duration of 14 days is adequate.

## 1. Introduction

The first observational evidence showing that the higher the sodium intake the higher the systemic inflammation—as measured by plasma c-reactive protein (CRP)—was published twelve years ago [1]. However, the magnitude of this association was only marginal (increase in CRP of 1.06 mg/L per 2.3 g/L of urinary sodium excretion) with the authors suggesting that dietary sodium consumption is unlikely to be an important modifiable risk factor for increased systemic inflammation [1]. However, since robust evidence suggest that higher sodium intake is associated with higher incidence of cardiovascular (CV) disease [2], it seems plausible that sodium induced inflammation might be another missing link, beyond blood pressure (BP) increase, leading to atherosclerosis, a per se inflammatory process [3].

Since then, several studies have been conducted in humans but mostly in animal models to test the hypothesis of sodium induced systemic inflammation [4]. In animal models, the majority of the available data, but not all [5,6], indicate that high sodium intake is associated with increased levels of circulating inflammatory biomarkers [7,8,9,10]. However, the evidence regarding the association of sodium intake and systemic inflammation in humans remain limited and controversial, in part due to the high heterogeneity and differences in the methodology applied e.g., regarding type of study (observation or intervention (acute or chronic) [11,12,13,14], the level of daily sodium intake tested [13,15], and the type of inflammatory biomarker that was evaluated [16,17]).

Of note, over the years this hypothesis has become even more intriguing because: (i) of the presence of a J-shape association between daily sodium intake and mortality in epidemiological studies [18,19,20], (ii) both very high and very low sodium intake are implicated in the pathogenesis of arterial damage [5,21,22,23,24], (iii) data derived from in vitro and in vivo experimental animal studies, as well as preliminary human studies, suggest an association between high sodium intake and autoimmune disease [25], and finally (iv) not only high, but also very low sodium intake seems to be proinflammatory [26,27]. However, given the fact that other factors (including certain types of foods, sedentary life, sleep apnea) [28,29] may have proinflammatory effects, the task to delineate and quantify the potential effects of sodium intake on systemic inflammation is quite complex.

In the present study we aimed to investigate the hypothesis that sodium intake induces systemic inflammatory response in humans in a dose response manner. To this end, we performed a systematic review and meta-analysis of all randomized studies comparing the effect of at least two different levels of dietary sodium intake on the magnitude of systematic inflammatory response, as described by predefined circulating inflammatory biomarkers. The primary endpoint of the meta-analysis was the magnitude of inflammatory response (i.e., the difference in the levels of each inflammatory biomarker) after the intervention with a high sodium diet (HSD) versus low sodium diet (LSD); the magnitude of BP response was defined as a secondary endpoint (i.e., the difference in BP).

## 2. Materials and Methods

### 2.1. Search Strategy

This systematic review and meta-analysis was performed according to PRISMA guidelines (Appendix A) [30]. A systematic search of potentially relevant studies was performed throughout April 2020 by two separate investigators (E.D. Basdeki & C. Tsirimiagkou) in PUBMED and SCOPUS databases. Search terms applied included: (“dietary sodium” OR “sodium intake” OR “sodium excretion” OR “urinary sodium” OR salt) AND (inflammation OR “inflammatory biomarkers” OR “inflammatory indices” OR “inflammatory cells” OR “white blood cells” OR “C-reactive protein” OR crp OR interleukin OR lymphocyte OR leukocytes OR il-5 OR il-6 OR il-10 OR il-12 OR il-23 OR il-17 OR cytokines OR “tumor necrosis factor” OR tnf OR tnf-a OR tnf-b OR cd4 OR cd8)). Articles were also identified from reference lists of relevant papers and hand search. Studies were limited to English language, human, and randomized controlled studies (RCTs). Disagreements were resolved by consensus with a senior author (A.D. Protogerou).

### 2.2. Inclusion & Exclusion Criteria

Eligible studies were full-text peer-reviewed articles in English that: 1. were RCTs with parallel-arm (different patients) or crossover (same patients) design, 2. conducted in males and/or females regardless of diseases (chronic or acute), 3. examined the effect of at least two different daily doses of sodium intake on circulating inflammatory biomarkers. The following exclusion criteria were applied: epidemiological studies, non-RCTs, animal studies, reviews, systematic reviews, meta-analyses, comments/letters.

### 2.3. Selection of Studies & Data Extraction

Two reviewers screened the available titles, abstracts and keywords from all of the available articles. Discrepancies were resolved after discussion. After agreement, full text screening was carried out. Both reviewers extracted independently qualitative and quantitative data from all included articles, concerning study design, population characteristics and data regarding primary endpoints from included studies where available. Authors of the included studies were contacted by e-mail to obtain additional details not reported in the published paper (i.e., mean and SD of difference regarding the variable of interest). The risk of bias was assessed using a Cochrane Collaboration’s tool for assessing risk of bias in randomized trials [31].

### 2.4. Statistical Analysis

Meta-analysis was performed using the Stata/SE 11 (Texas) software. Sensitivity analyses were performed to compensate for the observed methodological heterogeneity among the included studies. Meta-regression analysis was performed for assessing associations between the difference in (a) CRP or systolic/diastolic BP (SBP/DBP) and (b) sex, age, duration of intervention, and difference in sodium intake between the examined diet arms across the included studies. Sensitivity analyses were performed according to the design of the study (crossover or parallel), the mean age of the studied population, the average sodium intake in each arm, and by excluding studies with patients on hemodialysis. Mean values of subgroups were combined where feasible [32]. Median values were converted to mean values using appropriate formulas [33]. In the case of missing values regarding the mean (SD) of difference in the outcome of interest between the examined groups, these were calculated from the groups’ mean values using appropriate calculators [34]. The latter procedure was also implemented for paired comparisons in crossover studies as a rough approximation. Heterogeneity was tested using an I^2^ statistic. A value of I^2^ statistic >50% was considered to indicate significant heterogeneity between studies. When significant heterogeneity was present, a random-effects model of analysis was used; otherwise, a fixed-effects model of analysis was used. Publication bias was assessed by inspecting funnel plots, as well as Egger’s test (linear regression method) and Begg’s test (rank correlation method) [35,36]. Two-sided *p* values of <0.05 were considered significant.

## 3. Results

### 3.1. Number of Studies Screened and Selected—General Description

The PRISMA Checklist for the present systematic review and meta-analysis is presented in Appendix A. Three thousand six hundred and twenty-three (3623) studies were identified through a systematic search. The flow chart for study selection is shown in Figure 1. Eight studies [13,27,37,38,39,40,41,42] met the inclusion criteria for examining the effect of sodium intake on circulating inflammatory biomarkers and were included in the systematic review. Detailed descriptive data, as well as results for all the included studies are provided in Table 1 and Table 2.

All 8 eligible studies identified (published from 2005 to 2019) were predefined RCTs, 5 of which had a cross-over study design [13,27,39,41,42] and 3 had a parallel-arm study design [37,38,40] (Table 1). Each study had 2 sodium intervention periods (for cross-over design studies) or 2 sodium intervention groups (for parallel-arm design studies); no study with more than 2 sodium periods or groups was identified (Table 1). Half of the studies [27,38,39,42] used sodium capsules versus placebo as an add-on intervention to the diet; only 5 out of the 8 studies used 24-h urine collection as a sodium intake assessment method (Table 1).

Throughout this present text, the different levels/groups of daily sodium consumption in each study are quoted for simplicity as “HSD” and “LSD”. However, there was high heterogeneity between the identified studies regarding the level of daily sodium intake. LSD was highly variable, ranging from 115 to 6740 mg/day, and likewise HSD ranged from 1380 to 9240 mg/day (Table 1).

The examined inflammatory biomarkers were CRP in 6 out of 8 studies [13,27,37,38,41,42], TNF-a and IL-6 in 5 out of 8 studies [27,37,40,41,42], IL-8 in 2 out of 8 studies [39,41] and IL-10 [40], IL-12 [41], IL-1β [39], interferon-γ [42], and procalcitonin (PCT) [27], in 1 out of 8 studies for each biomarker.

All studies included small to moderate size populations (from 11 to 173 participants; male sex from 37% to 100%) with high heterogeneity (Table 1) regarding: (a) the type of populations investigated, (b) their age level (from 18 to 72 years old), and (c) duration of intervention (from 14 to 365 days; as well as one acute effect study) [13]. Finally, only 2 studies had a run-in period [38,42], and baseline sodium levels were described only in 3 studies [40,41,42] (Table 2); in only 1 out of 8 studies, salt sensitivity assessment was conducted [27]. No differences in the changes of the inflammation biomarkers (CRP, TNF-a, IL-6, PCT) were detected between salt sensitive and salt resistant individuals.

### 3.2. Systematic Review Results per Inflammatory Biomarker: Qualitative Description per Inflammatory Biomarker

Results on CRP (Table 1 and Table 2 and Figure 2): out of the 6 studies [13,27,37,38,41,42] investigating the effect of different levels of sodium intake on CRP, 1 was an acute effect study (evaluating the effect of sodium for 2 h, every 30 min), in which participants were asked to consume a meal that was low or high in sodium [13]. This acute effect study indicated no statistically significant changes in CRP results. Only 2 out of the 5 remaining studies showed statistically significant results [37,38].

Results of TNF-a (Table 1 and Table 2): 3 of the 5 studies [27,37,40,41,42] examining sodium intake and TNF-a showed statistically significant results [27,37,40]. In 2 of the studies [27,40], TNF-a levels after the LSD (460 mg/day [27] and 1800 mg/day [40]) intervention were significantly higher than that after the HSD (4600 mg/day [27] and 2800 mg/day [40]). In the third study [37], TNF-a decreased significantly only after LSD (6740 mg/day) compared to the baseline. No statistically significant results were found after HSD intervention (9240 mg/day) compared to baseline.

Results on IL-6 (Table 1 and Table 2): 3 out of the 5 studies [27,37,40,41,42] investigating sodium intake and IL-6 showed statistically significant results. In 1 study [41], IL-6 levels after the LSD (736 mg/day) intervention were significantly lower than that after the HSD (7452 mg/day). In 1 study [40], IL-6 levels after the LSD (1800 mg/day) intervention were significantly higher than those after the HSD (2800 mg/day). In the third study [37], IL-6 decreased significantly after LSD (6740 mg/day) compared to the baseline. No statistically significant results were found after HSD intervention (9240 mg/day) compared to baseline.

Results on IL-8 (Table 1 and Table 2): 1 of the 2 studies [39,41] investigating IL-8 found statistically significant results [39]; IL-8 levels after the intervention with LSD were significantly lower than that after the HSD intervention period.

Results on IL-10 (Table 1 and Table 2): only 1 study [40] examined sodium intake and IL-10, indicating that IL-10 levels after the intervention with LSD were significantly lower than that after the HSD intervention period.

Results on IL-12 (Table 1 and Table 2): only 1 study [41] investigated sodium intake and IL-12, but no statistically significant results were found.

Results on IL-1β (Table 1 and Table 2): only 1 study [39] investigated sodium intake and L-1β, indicating statistically significant results; IL-1β levels after the intervention with LSD were significantly lower than that after the HSD intervention period.

Results on Interferon-γ (Table 1 and Table 2): only 1 study [42] investigated sodium intake and interferon-γ, but no statistically significant results were found.

Results on PCT (Table 1 and Table 2): only 1 study [27] investigated sodium intake and PCT levels. PCT levels after the intervention with LSD were significantly higher than that after the HSD intervention period.

### 3.3. Meta-Analysis Results: Primary Endpoints

Out of the 8 studies, 6 [27,37,38,40,41,42] were included in the meta-analysis (Figure 1). One study [39] was not included since it investigated inflammatory biomarkers not measured in any of the rest of the studies. A second study [13] was excluded because it was the only acute effect study (2 h duration of intervention period); therefore, the results were not comparable to the rest of the long-duration studies (14 to 180 days). The assessment of the risk of bias is presented in Appendix A.

Five studies provided data on the CRP difference between HSD versus LSD and were included in the meta-analysis [27,37,38,41,42] (*n* = 273, weighted age 47.7 ± 8.4 years, men 47%, hypertension 23.4%). HSD versus LSD resulted in a pooled difference in CRP values (HSD–LSD) of 0.1 (95% confidence intervals [CI] −0.3, 0.4) mg/L (Figure 3). No publication bias was identified (all *p* = NS, Begg’s funnel plot is presented in Appendix A). Meta-regression analysis did not reveal any significant associations between the difference in CRP and male percentage, mean age, duration of intervention and difference in sodium intake between the examined diet arms across the included studies (all *p* = NS). In a sensitivity analysis excluding the study of Telini et al. [37] (which included patients on hemodialysis and had parallel arm design), the pooled difference in CRP values was similar at 0.1 (−0.3, 0.4) mg/L. In another sensitivity analysis including only the 3 studies [27,41,42] with crossover design, the pooled estimate was −0.1 (−0.4, 0.3) mg/L. By selecting the 2 studies with average sodium intake at the lowest (<1000 mg/day) and highest (>4500 mg/day) range for each arm [27,41], the pooled difference was −0.1 (−0.4, 0.3) mg/L. Three studies had populations with an average age <50 years [27,38,41] and the pooled difference calculated from these was 0.2 (−0.4, 0.7) mg/L.

Four studies provided data on the TNF-difference between HSD versus LSD and were included in the meta-analysis [27,37,40,42] (*n* = 264, weighted age 67.1 ± 8.7 years, men 60%, hypertension 45.8%). They showed that the HSD versus the LSD resulted in a pooled difference in TNF-a values (HSD-LSD) of −0.7 (−5.0, 3.6) pg/mL (Figure 4). No publication bias was identified (all *p* = NS, Begg’s funnel plot is presented in Appendix A). Meta-regression analysis did not reveal any significant associations between the difference in TNF-a and male percentage, mean age, duration of intervention and difference in sodium intake between the examined diet arms across the included studies (all *p* = NS). In a sensitivity analysis excluding the study of Telini et al. [37] (which included patients on hemodialysis and had a parallel arm design), the pooled difference in TNF-a values showed a pooled difference of −1.3 (−3.5, 0.8) pg/mL. Three studies had populations with an average age >55 years [37,40,42] and the pooled difference calculated from these was 2.8 (−7.8, 13.5) pg/mL.

Four studies provided data on the IL-6 difference between HSD versus LSD and were included in the meta-analysis [27,37,40,42] (*n* = 264, weighted age 67.1 ± 8.7 years, men 60%, hypertension 45.8%). They showed that the HSD versus the LSD resulted in a pooled difference in IL-6 values (HSD-LSD) of −1.1 (−3.3, 1.1) pg/mL (Figure 5). No publication bias was identified (all *p* = NS, Begg’s funnel plot is presented in Appendix A). Meta-regression analysis did not reveal any significant associations between the difference in IL-6 and male percentage, mean age, duration of intervention and difference in sodium intake between the examined diet arms across the included studies (all *p* = NS). In a sensitivity analysis excluding the study of Telini et al. [37], the pooled difference in IL-6 was −2.4 (−4.9, 0.2) pg/mL. Three studies had populations with an average age >55 years [37,40,42] and the pooled difference calculated from these was −1.8 (−5.3, 1.7) pg/mL.

### 3.4. Meta-Analysis Results: Secondary End-Points—Blood Pressure

Five studies [27,37,40,41,42] (*n* = 275, weighted age 65.5 ± 11.5 years, men 61%, hypertension 44%) reported the effect of HSD vs. LSD on SBP/DBP difference with a pooled estimate of 4.5 (−1.4, 10.4)/2.2 (−0.1, 4.4) mmHg (Figure 6). No publication bias was identified (all *p* = ns, Begg’s funnel plots are presented in Appendix A). Meta-regression analysis showed a higher difference in SBP with a shorter duration of the intervention (in days) across the included studies, whereas this was not evident for DBP (*p* = 0.03/ns respectively; Appendix A). No significant associations were observed between the difference in SBP/DBP and male percentage, mean age and difference in sodium intake between the examined diet arms across the included studies (all *p* = ns). The following sensitivity analyses were conducted: the first analysis excluded the study of Telini et al. [37] and showed a pooled SBP/DBP difference of 5.0 (−1.8 to 11.8)/2.4 (0.1 to 4.7) mm Hg. The second analysis included only the 3 studies [27,41,42] with crossover design (same patients) and showed a pooled SBP/DBP difference of 8.1 (4.1,12.2)/2.6 (−0.3, 5.5) mm Hg. The third analysis included studies with an average age of the population >55 years [37,40,42] and showed a pooled SBP/DBP difference of 2.2 (−5.6, 10.1)/2.2 (−1.0, 5.4) mm Hg. The fourth analysis included the 2 studies with average sodium intake at the lowest (<1000 mg/day) and highest (>4500 mg/day) range for each arm [27,41], and showed a pooled SBP/DBP difference of 7.7 (3.4, 12.0)/2.1 (−1.0, 5.3) mm Hg.

## 4. Discussion

In this present systematic review, we identified eight RCT studies that compared the effect of two different levels of dietary sodium intake on the magnitude of systematic inflammatory response, by assessing overall nine different circulating biomarkers (CRP, TNF-a, IL-1β, IL-6, IL-8, IL-10, IL-12, interferon-γ and hs-PCT). The meta-analysis of studies was feasible only for three inflammatory biomarkers (CRP, TNF-a, IL-6); all three of them showed non-significant differences in the primary endpoint i.e., the difference in circulating inflammatory biomarkers after HSD versus LSD.

The qualitative description of the included studies revealed inconsistent results for all biomarkers, as well as major methodological limitations (e.g., small sample size, poor sodium intake assessment methods) and high heterogeneity regarding major methodological traits (e.g., sodium doses, age, underlying diseases). Of note, there was extremely high heterogeneity and variability regarding the level of daily sodium intake that renders the used terms “LSD” and “HSD” relative and valid mostly for within each study comparison. This major limitation should be taken into account for the interpretation of meta-analysis results.

CRP was the inflammatory biomarker most commonly studied in the RCTs studies (overall six studies [13,27,37,38,41,42]; five included in the meta-analysis), however the overall negative result of the meta-analysis is limited by numerous methodological limitations, as previously discussed. Only two of the studies examined high sensitivity CRP (hs-CRP) [27,38], and only one study investigated more than 40 participants (the higher sample size was 171 [38]), whereas the overall sample size included in the metanalysis was 273. Actually, the most convincing data regarding a positive inflammatory response in HSD were derived from a single study [38], which is the only one that satisfied all of the following necessary methodological characteristics, i.e., evaluated population without low- or high-grade inflammation, had adequate sample size and methodology for sodium intake assessment and, most importantly, compared two “reasonable” levels of daily LSD and HSD (i.e., LSD of 1840 mg/day versus HSD of 3680 mg/day). On the contrary, other studies examined healthy populations either with a very small sample size (*n* = 11) [41], or examined a small number of diseased populations that exhibited at least low (if not high) grade inflammation (e.g., chronic kidney disease) [37,42]. Moreover, two studies [27,41] evaluated the effect of very low doses of daily sodium intake (460 to 736 mg/day) versus high doses of daily sodium intake; given the fact that both very low and high doses of sodium may be proinflammatory, the conclusions from these studies are not easy to interpret [26,27]; therefore, this may have limited the detection of any significant differences in inflammatory response. The only acute phase study that evaluated inflammatory response after a few hours of sodium intake used extremely low doses versus the usual dose of sodium intake [13].

TNF-a and IL-6 were examined in five studies; the same four of which were included in the respective meta-analysis with no overall significant effects for both biomarkers. All the previous discussed limitations are also present in these four studies. One study [27], that evaluated the effect of a very low sodium intake (i.e., LSD: 460 mg sodium/day versus HSD: 4600 mg sodium/day) showed that LSD significantly increased TNF-a levels versus HSD; however, as already discussed, it has been suggested that a very low sodium diet may also be proinflammatory [26,27]. On the other hand, two of the studies [40,42] that evaluated close to the internationally recommended LSD level (i.e., LSD: 1800 mg sodium/day and LSD: 1700 mg sodium/day) versus reasonable HSD, showed conflicting results.

All of the other identified inflammatory biomarkers (IL-1β, IL-8, IL-10, IL-12, interferon-γ and hs-PCT) have been barely evaluated since they were all examined just once [27,39,40,41,42], except for IL-8, which was examined in two different studies [39,41]; however, only one found significant results [39]. Overall, all of the above studies but one [41] were conducted in non-healthy subjects (i.e., asthmatic patients, chronic kidney disease, hypertensives, heart failure patients), with a limited number of participants: less than 33 in each included study, with the exception of one study that was conducted in 173 heart failure patients and evaluated IL-10 levels [40]. Although three of the studies indicated statistically significant results [27,39,40] regarding IL-8, IL-1β, IL-10 and hs-PCT, safe conclusions cannot be drawn given the described limitations above.

Given the fact that the association between daily sodium intake and BP levels is very well established [2,43], we included in the present meta-analysis—as a secondary end point—the change of BP levels, between HSD versus LSD, using the extracted data from the very same studies that were used to evaluate inflammatory biomarkers. The fact that the principal BP meta-analysis showed marginally no significant associations between sodium and SBP/DBP does verify that the previously described methodological limitations of the included studies are important and may have indeed limited our ability to identify an association between sodium intake and systematic inflammation biomarkers.

The present systematic review and meta-analysis has some major limitations, mainly due to the methodological heterogeneity of the included studies. First, the available studies have been conducted in populations with different characteristics and types of chronic diseases. The existence of chronic diseases might have also influenced participants’ levels of inflammation. Second, there was heterogeneity in the duration of intervention and the intake of sodium among the included studies. Third, the number of the studies included was small, as this topic is understudied. Four, several other undetermined factors (diet, quality of sleep, exercise) might have confounded the final results. However, the abovementioned limitations were, at least partly, compensated by sensitivity and stratified analyses. In addition, the significant effect of the increased sodium intake on BP levels is reassuring for the use of validated methodology in the included studies.

Beside the negative results, the present systematic review helped us to critically revise all the literature and to identify major issues that must be addressed in future RCT efforts to address this hypothesis. Based on previously discussed data, we recommend that such RCT should have three arms (very low, “normal” and high) of daily sodium intake, with sample sizes of more than 100 participants per arm, and should focus on healthy and young participants to avoid the confounding effect of aging and comorbidities. A short study duration of 14 days with a seven days run-in period has been proven to be sufficient, and seems to be more adequate than longer studies, in order to accomplish maximal adherence to the study diets which, however, should be very closely monitored and verified with repeated 24-h urine selections. Most importantly, the above studies were all focused on circulatory biomarkers of inflammation and cannot provide evidence of tissue response or other types of inflammatory (e.g., cellular) response. Therefore, future studies should not only use state-of-the-art methodology to assess circulating biomarkers of inflammation, but should also provide evidence at a cellular level.

## 5. Conclusions

To conclude, this present study failed to verify the hypothesis that sodium intake induces systemic inflammatory response in humans in a dose response manner. However, due to the aforementioned major limitations, the negative results of the meta-analysis are neither convincing nor provide a definite response to the hypothesis. Future better designed RCTs addressing all of the issues that were discussed above are needed, since a potentially weak association between dietary sodium intake and systemic inflammation, as suggested by Fogarty et al. in the very first publication [1] on the topic, needs a very carefully designed trial to be properly quantified.

## Figures and Tables

**Figure 1 nutrients-13-02632-f001:**
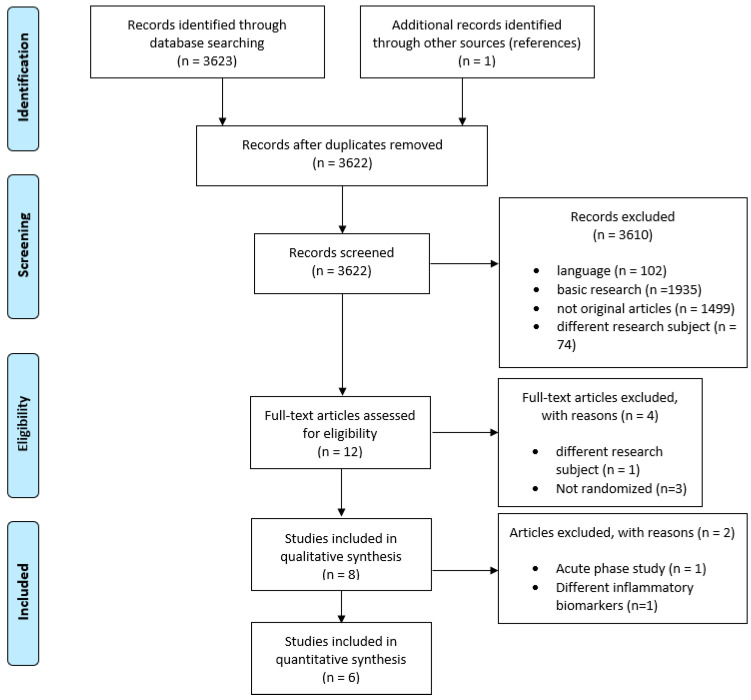
Identification and selection of the eligible studies according to the PRISMA criteria.

**Figure 2 nutrients-13-02632-f002:**
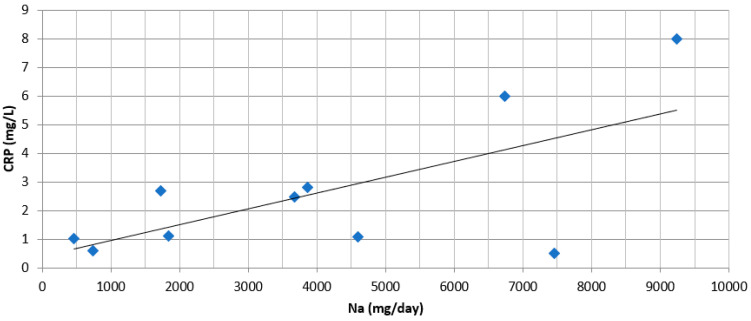
Scatter dot plot of average daily sodium intake (mg/day) and CRP plasma levels (mg/L) per study group. Data from 5 available randomized studies [27,37,38,41,42], after excluding 1 study [13], which evaluated the acute effect (after one meal, every 30 min, for 2 h) of sodium intake. Range of duration of sodium intervention was 14 to 112 days. r = 0.663; *p* = 0.037.

**Figure 3 nutrients-13-02632-f003:**
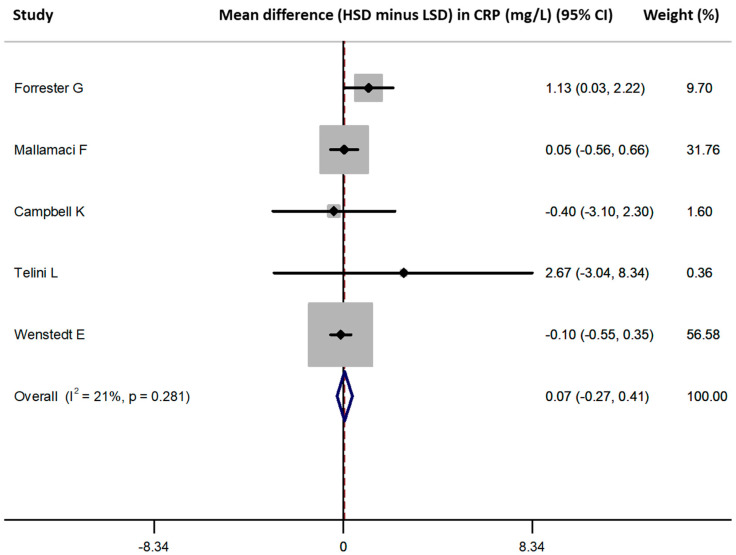
Forest plot of the mean difference in CRP levels for high sodium dose (HSD) versus low sodium dose (LSD), as defined per study [27,37,38,41,42].

**Figure 4 nutrients-13-02632-f004:**
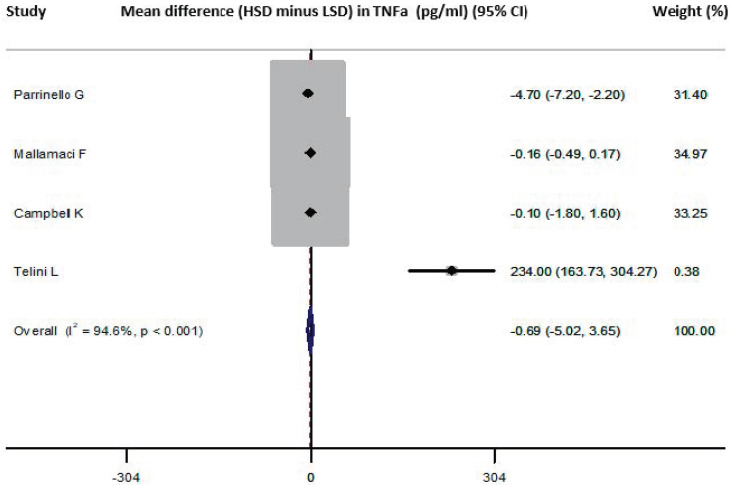
Forest plot of the mean difference in TNF-a levels for high sodium dose (HSD) versus low sodium dose (LSD) as defined per study [27,37,40,42].

**Figure 5 nutrients-13-02632-f005:**
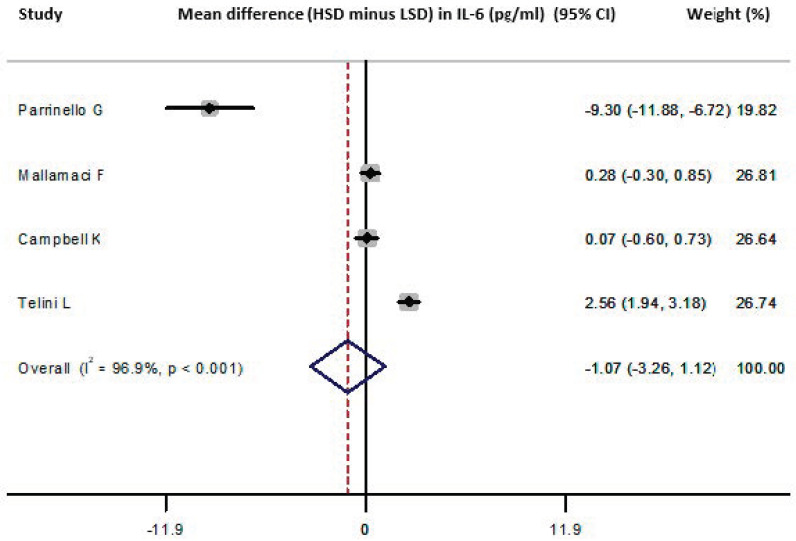
Forest plot of the mean difference in IL-6 levels for high sodium dose (HSD) versus low sodium dose (LSD) as defined per study [27,37,40,42].

**Figure 6 nutrients-13-02632-f006:**
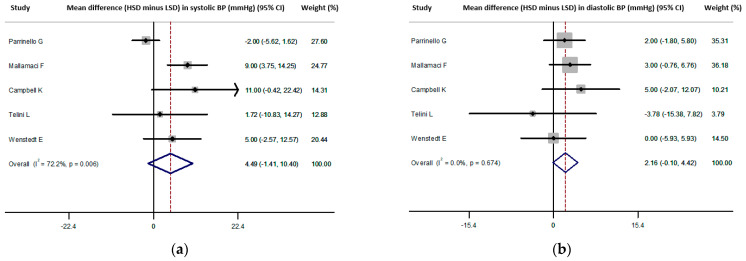
(**a**) Forest plot of the mean difference in systolic blood pressure (SBP) levels for high sodium dose (HSD) versus low sodium dose (LSD) as defined per study [27,37,40,41,42]. (**b**) Forest plot of the mean difference in diastolic blood pressure (DBP) levels for high sodium dose (HSD) versus low sodium dose (LSD) as defined per study [27,37,40,41,42].

**Table 1 nutrients-13-02632-t001:** Descriptive characteristics of all of the selected studies (*n* = 8).

Author (Year)	Study Design	Population Description	*N*	Sex (% Male)	Age (Mean ± SD or Range)	Run-in Period (Days) *	Duration (Days)	Intervention (mg Sodium/d)	Sodium Intake Method	Sodium Assessment Method	Inflammatory Biomarkers at Baseline (Mean ± SD)	Salt Sensitivity Assessment
Mickleborough T. (2005) [39]	randomized, db, cross-over, placebo controlled	Treated mild asthma	24	62.5	24 ± 1.8	no	14	LSD: 1500 HSD: 5500	diet + sodium capsules or placebo	24 hU	IL-1β: n/a ** IL-8: n/a **	no
Parrinello G. (2009) [40]	randomized, db, 2 parallel arms	HF patients	173	60.7	72.5 ± 7	no	180	LSD: 1800 HSD: 2800	diet	dietary methods	TNF-a: 19.1 ± 8.6 (LSD) 17.8 ± 9 (HSD) IL-6: 20.8 ± 6.9 (LSD) 21.3 ± 12.5 (HSD) IL-10: 68.7 ± 5.6 (LSD) 62.8 ± 5.4 (HSD)	no
Forrester G. (2010) [38]	randomized, db, placebo controlled, 2 parallel arms	Asthma & measurable bronchial reactivity to methacholine	171	37.5	44.2 ± 12.2	7	42	LSD: 1840 HSD: 3680	diet + sodium capsules or placebo	24 hU	*hs-CRP:* n/a	no
Mallamaci F. (2013) [27]	randomized, sb, cross-over, placebo controlled	mild-to-moderate HTN, CVD free, no anti-HTN drugs	32	72	48 ± 9	no	14	LSD: 460 HSD: 4600	diet + sodium capsules or placebo	24 hU	hs-CRP: n/a TNF-a: n/a IL-6: n/a hs-PCT: n/a	yes
Dickinson K. (2014) [13]	randomized, cross-over	NT, ΒΜΙ: 18–27 Kg/m^2^	16	43.75	18–70	no	0.1	LSD: 115 HSD: 1495	diet	N/A	*CRP:* n/a	no
Campbell K. (2014) [42]	randomized, db, cross-over, placebo controlled	(P)HT, Stage III & IV CKD	20	75	68.5 ± 11	7	14	LSD: 1380 to 1840 HSD: 1380 to 1840 plus 2760	diet + sodium capsules or placebo	24 hU	CRP: 3.6 ± 3.4 TNF-a: n/a IL-6: n/a Interferon-γ: n/a	no
Telini L. (2014) [37]	randomized controlled study, 2 parallel arms	CKD—hemodialysis for at least 90 days	39	38.5	57.9 ± 12.8	no	112	LSD: habitual diet minus 2000 HSD: habitual diet	diet	dietary methods	CRP: 11.3± 3.9 (LSD) 11.8± 4.8 (HSD) TNF-a: 694.7 ± 101 (LSD) 651± 96.5 (HSD) IL-6: 5.4± 0.7 (LSD) 5.7± 0.6 (HSD)	no
Wenstedt E. (2019) [41]	randomized, cross-over	healthy, non-smoking	11	100	28 ± 5	no	14	LSD: <1200 HSD: >4800	diet	24 hU	CRP: n/a TNF-a: n/a ** IL-6: n/a ** IL-8: n/a ** IL-12: n/a **	no

ABBREVATIONS: sb: single blind; db: double—blind; HT: hypertensives; NT: normotensives; PHT: pro-hypertensives; prosp: prospective; 24 hU: 24-h urine collection; HF: heart failure; CKD: chronic kidney disease; CVD: cardiovascular disease; LSD: low sodium diet; HSD: high sodium diet; BMI: body mass index; HS: high sodium; LS: low sodium; N/A: Not available; PCT: procalcitonin; SS: salt sensitivity; * During the run in period all participants received standard daily Na intake diet; n/a **:only diagrams were provided, accurate data not available.

**Table 2 nutrients-13-02632-t002:** Brief qualitative description of results, primary and secondary end points, per study (*n* = 8).

Author (Year)	Duration (Days)	Intervention (mg Sodium/d)	Primary End Point									Secondary End Point
			CRP	TNF-a	IL-6	IL-8	IL-10	IL-12	IL-1β	Interferon-γ	hs-PCT	BP Change
Mickleborough T. (2005) [39]	14	Baseline: n/a LSD: 1500 HSD: 5500	-	-	-	LSD vs. HSD: ↓	-	-	LSD vs. HSD: ↓	-	-	ns
Parrinello G. (2009) [40]	180 **	Baseline: 2600 LSD: 1800 HSD: 2800	-	LSD vs. HSD: ↑	LSD vs. HSD: ↑	-	LSD vs. HSD: ↓	-	-	-	-	ns
Forrester G. (2010) [38]	42	Baseline: n/a LSD: 1840 HSD: 3680	LSD vs. HSD: ↓	-	-	-	-	-	-	-	-	n/a
Mallamaci F. (2013) [27]	14	Baseline: n/a LSD: 460 HSD: 4600	LSD vs. HSD: ns	LSD vs. HSD: ↑	LSD vs. HSD: ns	-	-	-		-	LSD vs. HSD: ↑	After HSD: ↑24 h & night-time & daytime BP
Dickinson K. (2014) [13]	0.1	Baseline: n/a LSD: 115 HSD: 1495	LSD vs. HSD: ns	-	-	-	-	-	-	-	-	ns
Campbell K. (2014) [42]	14	Baseline: 3200 LSD: 1725 HSD: 3864	LSD vs. HSD: ns	LSD vs. HSD: ns	LSD vs. HSD: ns	-	-	-	-	LSD vs. HSD: ns	-	Peripheral SBP & DBP & central SPB: ↓ in LSD vs. HSD
Telini L. (2014) [37]	112	Baseline: n/a LSD: 6740 HSD: 9240	* LSD: ↓ * HSD: ns	* LSD: ↓ * HSD: ns	* LSD: ↓ * HSD: ns	-	-	-	-	-	-	ns
Wenstedt E. (2019) [41]	14	Baseline: 4000 LSD: 736 HSD: 7452	LSD vs. HSD: ns	LSD vs. HSD: ns	LSD vs. HSD: ↓	LSD vs. HSD: ns	-	LSD vs. HSD: ns	-	-	-	SBP: ↑ in HSD

ABBREVATIONS: BP: blood pressure; SBP: systolic blood pressure; DBP: diastolic blood pressure; MAP: mean arterial pressure; LSD: low sodium diet; HSD: high sodium diet; n/a: not available; ns: non-significant; vs: versus; ns: no statistically significant change; HS: high sodium; LS: low sodium; N/A: not available; PCT: procalcitonin; ↑ or ↓: statistically significant difference (higher or lower) between LSD & HSD. * Indicates statistically significant differences between LSD or HSD and baseline. ** The study evaluated outcome both at 180 and 365 days; the presented data correspond to the 180 days intervention in order to minimize the duration gap with the rest of the studies which presented maximum duration 112 days. Level of statistical significance *p* < 0.05 for differences presented either between LSD and HSD or between baseline and modified sodium diet.

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
