# Peer review of "Does Sodium Intake Induce Systemic Inflammatory Response? A Systematic Review and Meta-Analysis of Randomized Studies in Humans"

_nutrients, 2021, doi:10.3390/nu13082632_

Round 1

Reviewer 1 Report

Dear authors,

The introduction and the discussion should include information about the relevance of other habits that can be inflammatory like other life style aspect (diet, exercise, sleep, work...) and not only sodium intake. Those habits can be altering the results.

The numbers in Figure 1 are not correct in the last step. 8 - 3 is 5, not 6.

The methods are clearly described but I do not see reliable to mix in a meta-analysis samples with different kind of chronic diseases that can influence also their level of inflammation. Also the duration of the intervention and the intake of sodium is not similar in most cases. We can be given wrong advice for the readers and for many people because of mixing very different samples with different kinds of intervention duration and very different sodium intake.

Have you done sensitivity analysis for each meta-analysis? 

Figure 3 meta-analysis should not use random effects because i2 is low. The same for the others with low heterogeneity.

The forestplots do not clearly show where is favor for high sodium or where is favor for low sodium.

Kind regards

Author Response

Reviewer 1

Dear authors,

The introduction and the discussion should include information about the relevance of other habits that can be inflammatory like other life style aspect (diet, exercise, sleep, work...) and not only sodium intake. Those habits can be altering the results.

We agree with the comment. There are several undetermined factors that may also be proinflammatory (including types of foods, sedentary life, sleep apnea and more). In the revised manuscript we have added the following sentences:

Introduction section, line 70: “However, given the fact that other factors (including certain types of foods, sedentary life, sleep apnea) [28, 29] may have proinflammatory effect the task to delineate and quantify the potential effect of sodium intake on systemic inflammation is quite complex.

Discussion section, line 387: “Third, the number of the studies included was small, as this topic is understudied. Four, several other undetermined factors (diet, quality of sleep, exercise) might have confounded the final results.

The numbers in Figure 1 are not correct in the last step. 8 - 3 is 5, not 6.

We would like to apologize for this and thank the reviewer for his/her comment. The mistake in figure 1 has been corrected. 

The methods are clearly described but I do not see reliable to mix in a meta-analysis samples with different kind of chronic diseases that can influence also their level of inflammation. Also the duration of the intervention and the intake of sodium is not similar in most cases. We can be given wrong advice for the readers and for many people because of mixing very different samples with different kinds of intervention duration and very different sodium intake.

We understand the reviewer’s concern regarding the limitations of our study. However, we would like to address the importance of this work. It is the first attempt to systematically review the current evidence. Indeed, there are some limitations in the included studies, i.e. heterogeneity in the characteristics of the included populations, as well as in type and duration of intervention, overlap in sodium ranges, restricted number of studies. However, sensitivity analyses and meta-regression analysis were performed to account for the observed heterogeneity and our main conclusion is that future well-designed studies are needed.

In respect to the reviewer’s comment, we have now added a paragraph mentioning these limitations in the end of the Discussion section. The added paragraph is:

-Discussion section (line 382)

The present systematic review and meta-analysis has some major limitations, mainly due to the methodological heterogeneity of the included studies. First, the available studies have been conducted in populations with different characteristics and types of chronic diseases. The existence of chronic diseases might have also influenced participants’ level of inflammation. Second, there was heterogeneity in the duration of intervention and the intake of sodium among the included studies. Third, the number of the studies included was small, as this topic is understudied. Four, several other undetermined factors (diet, quality of sleep, exercise) might have confounded the final results. However, the abovementioned limitations were, at least partly, compensated by sensitivity and stratified analyses. In addition, the significant effect of the increased sodium intake on BP levels is reassuring for the use of validated methodology in the included studies.

Have you done sensitivity analysis for each meta-analysis? 

The following text has been added in the Methods (statistical analysis section, line 120):

“Sensitivity analyses were performed according to the design of the study (crossover or parallel), the mean age of the studied population, the average sodium intake in each arm, and by excluding studies with patients on hemodialysis.”

The following text has been added in the Results:

Line 245:

“By selecting the 2 studies with average sodium intake at the lowest (<1000 mg/day) and highest (>4500 mg/day) range for each arm [27,41], the pooled difference was -0.1 (-0.4, 0.3) mg/L. Three studies had populations with average age <50 years [27,38,41] and the pooled difference calculated from these was 0.2 (-0.4, 0.7) mg/L.”

Line 264:

“Three studies had populations with average age >55 years [37, 40, 42] and the pooled difference calculated from these was 2.8 (-7.8, 13.5) pg/mL.”

Line 280:

“In sensitivity analysis excluding the study of Telini et al [37] the pooled difference in IL-6 was -2.4 (-4.9, 0.2) pg/mL. Three studies had populations with average age >55 years [35, 40, 42] and the pooled difference calculated from these was -1.8 (-5.3, 1.7) pg/mL.”

Line 300:

“The following sensitivity analyses were conducted: The first analysis excluded the study of Telini et al [37] and showed a pooled SBP/DBP difference of 5.0 (-1.8 to 11.8)/2.4 (0.1 to 4.7) mm Hg. The second analysis included only the 3 studies [27, 41, 42] with crossover design (same patients) and showed a pooled SBP/DBP difference of 8.1 (4.1, to 12.2)/2.6 (-0.3, to 5.5) mm Hg. The third analysis included studies with average age of the population > 55 years [35,38,40] and showed a pooled SBP/DBP difference of 2.2 (-5.6, 10.1)/2.2 (-1.0, 5.4) mm Hg. The fourth analysis included the 2 studies with average sodium intake at the lowest (<1000 mg/day) and highest (>4500 mg/day) range for each arm [27, 41], and showed a pooled SBP/DBP difference of 7.7 (3.4, 12.0)/2.1 (-1.0, 5.3) mm Hg.”

Figure 3 meta-analysis should not use random effects because i2 is low. The same for the others with low heterogeneity.

Thank you for this comment.

  1. The following text in the Methods (line 115):

“Random-effects meta-analysis was performed using the Stata/ SE 11 (Texas) software…”

has been changed as follows (line 115):

“Meta-analysis was performed using the Stata/ SE 11 (Texas) software.”

  1. The following text in the Methods section (line 131):

“Heterogeneity was tested using I2 statistics.”

has been changed as follows (line 128):

“Heterogeneity was tested using I2 statistic. A value of I2 statistic >50% was considered to indicate significant heterogeneity between studies. When significant heterogeneity was present, a random-effects model of analysis was used; otherwise, a fixed-effects model of analysis was used.

Figure 3 has been reformatted as follows:

Updated results have been provided in the text (Abstract, line 32 & Results section, line 237).

Figure 6 regarding the difference in DBP did not change at all by performing a fixed effect model analysis.

The forest plots do not clearly show where is favor for high sodium or where is favor for low sodium.

This is now clearly stated in the heading of each figure as follows:

Mean difference (HSD minus LSD) in (examined variable)

Reviewer 2 Report

General comment: Review article entitled “Does sodium intake induce systemic inflammatory response? Α systemic review and meta-analysis of randomized studies in humans” is a well-organized study, with sufficient methodology and discussion of the results. Some minor corrections are required for the improvement of the manuscript.

Abstract: The Abstract is well written and adequately presents the the aim and the basic results of the study.

Introduction: The introduction section is well-written and covers the importance to further understand the role of sodium on inflammatory biomarkers.

Materials and Methods:  The materials and methods are adequately presented.

Results: The results of the study are analytically presented. Tables and Figures are adequate explain the findings of the study.

Discussion: The results of study are sufficiently discussed.

-Please underline possible limitations of the study in a short paragraph.

References: The references used by the authors cover adequately the relative scientific field and the aims of the study.

This article is one of the litle that I performent very minor comments, because I observed that is is very well written in all the sessions. It is an article with adequte methodology for systematic reviews according to PRISMA, sufficient metanalysis process and sufficient presentation of the results with taables and figures and discussion of them. So I have not any comment for correction, except of addition of paragraph for possible limitations. In that point we could add in the comments a parenthesis (possible small number of reviewed articles, possible alternative added biomarkers etc.).

Author Response

Reviewer 2

General comment: Review article entitled “Does sodium intake induce systemic inflammatory response? Α systemic review and meta-analysis of randomized studies in humans” is a well-organized study, with sufficient methodology and discussion of the results. Some minor corrections are required for the improvement of the manuscript. Abstract: The Abstract is well written and adequately presents the the aim and the basic results of the study. Introduction: The introduction section is well-written and covers the importance to further understand the role of sodium on inflammatory biomarkers. Materials and Methods:  The materials and methods are adequately presented. Results: The results of the study are analytically presented. Tables and Figures are adequate explain the findings of the study. Discussion: The results of study are sufficiently discussed. Please underline possible limitations of the study in a short paragraph. References: The references used by the authors cover adequately the relative scientific field and the aims of the study. This article is one of the litle that I performent very minor comments, because I observed that is is very well written in all the sessions. It is an article with adequte methodology for systematic reviews according to PRISMA, sufficient metanalysis process and sufficient presentation of the results with taables and figures and discussion of them. So I have not any comment for correction, except of addition of paragraph for possible limitations. In that point we could add in the comments a parenthesis (possible small number of reviewed articles, possible alternative added biomarkers etc.).

We want to thank the reviewer for his/her comments about our work. We have taken into consideration the comment about the limitations of this study, and we added a paragraph analyzing these limitations in the end of the discussion section. The added paragraph is:

-Discussion section (line 382)

The present systematic review and meta-analysis has some major limitations, mainly due to the methodological heterogeneity of the included studies. First, the available studies have been conducted in populations with different characteristics and types of chronic diseases. The existence of chronic diseases might have also influenced participants’ level of inflammation. Second, there was heterogeneity in the duration of intervention and the intake of sodium among the included studies. Third, the number of the studies included was small, as this topic is understudied. Four, several other undetermined factors (diet, quality of sleep, exercise) might have confounded the final results. However, the abovementioned limitations were, at least partly, compensated by sensitivity and stratified analyses. In addition, the significant effect of the increased sodium intake on BP levels is reassuring for the use of validated methodology in the included studies.

Reviewer 3 Report

The authors have performed meta-analysis and have interpreted the results very well. There are some points which when addressed will make the review more compelling.

The ranges of LSD and HSD share common ranges. How do the authors take that into account. Rather than using these terms, the authors can consider to use the salt ranges. Would the analyses make more sense, if there is an accountability of dose in the analysis?

How would the authors interpret BP changes due to sodium level change. There seems to be conflicting data in the literature. Please elaborate on that.

How do the authors correct for the age ranges used in different studies?

The cytokine level changes demonstrate heterogeneous trend in different publications. How do authors account for it? How do the authors explain discrepancy?

What do the rodents study confer? Does those results is in line with any of the published literature that has been discussed?

Please write few lines on the prospective experiments that would address the current issue. How would the trial look like? How does the meta-analysis help in designing new experiments that would answer the question?

Author Response

Reviewer 3

The authors have performed meta-analysis and have interpreted the results very well. There are some points which when addressed will make the review more compelling. The ranges of LSD and HSD share common ranges. How do the authors take that into account? Rather than using these terms, the authors can consider to use the salt ranges. Would the analyses make more sense, if there is an accountability of dose in the analysis?

This is an important point and represents one of the major limitations of this analysis mentioned in the limitations paragraph in the Discussion. Actually, only 2 studies had average sodium intake at the lowest (<1000 mg/day) and highest (>4500 mg/day) range for each arm. Sensitivity analyses were performed, wherever feasible, as follows:

Regarding the difference in CRP (Results section, line 245):

“By selecting the 2 studies with average sodium intake at the lowest (<1000 mg/day) and highest (>4500 mg/day) range for each arm [27, 41], the pooled difference was -0.1 (-0.4, 0.3) mg/L.”

Regarding the difference in SBP/DBP (Results section, line 307):

“The fourth analysis included the 2 studies with average sodium intake at the lowest (<1000 mg/day) and highest (>4500 mg/day) range for each arm [27, 41], and showed a pooled SBP/DBP difference of 7.7 (3.4, 12.0)/2.1 (-1.0, 5.3) mm Hg.”

How would the authors interpret BP changes due to sodium level change. There seems to be conflicting data in the literature. Please elaborate on that.

We would like to thank the reviewer for his/her comment, since we are given the chance to explain this part of the study more thoroughly. Reduction in sodium intake leads to reduction in blood pressure levels, in adults with or without hypertension [1, 2], even if the intake level still remains above recommendations [3, 4]. As far as we know, there is no available study in the literature reporting a J-shaped relationship between sodium intake and blood pressure levels.

  1. Huang L, Trieu K, Yoshimura S, Neal B, Woodward M, Campbell NRC, et al. Effect of dose and duration of reduction in dietary sodium on blood pressure levels: systematic review and meta-analysis of randomised trials. Bmj. 2020;368:m315.
  2. Filippou CD, Tsioufis CP, Thomopoulos CG, Mihas CC, Dimitriadis KS, Sotiropoulou LI, et al. Dietary Approaches to Stop Hypertension (DASH) Diet and Blood Pressure Reduction in Adults with and without Hypertension: A Systematic Review and Meta-Analysis of Randomized Controlled Trials. Advances in nutrition. 2020;11(5):1150-60.
  3. He, F.J., J. Li, and G.A. Macgregor, Effect of longer term modest salt reduction on blood pressure: Cochrane systematic review and meta-analysis of randomised trials. BMJ, 2013. 346: p. f1325.
  4. Graudal, N.A., T. Hubeck-Graudal, and G. Jurgens, Effects of low sodium diet versus high sodium diet on blood pressure, renin, aldosterone, catecholamines, cholesterol, and triglyceride. Cochrane Database Syst Rev, 2017. 4: p. CD004022

How do the authors correct for the age ranges used in different studies?

Relevant sensitivity analyses according to the mean age of the studied populations have been performed. The following text has been added in the Results:

Regarding the difference in CRP (line 248):

“Three studies had populations with average age <50 years [27, 38, 41] and the pooled difference calculated from these was 0.2 (-0.4, 0.7) mg/L.”

Regarding the difference in TNF-a (line 264):

“Three studies had populations with average age >55 years [37,40, 42] and the pooled difference calculated from these was 2.8 (-7.8, 13.5) pg/mL.”

Regarding the difference in IL-6 (line 282):

“Three studies had populations with average age >55 years [37, 40, 42] and the pooled difference calculated from these was -1.8 (-5.3, 1.7) pg/mL.”

Regarding the difference in SBP/DBP (line 305):

“The third analysis included studies with average age of the population > 55 years [37, 40, 42] and showed a pooled SBP/DBP difference of 2.2 (-5.6, 10.1)/2.2 (-1.0, 5.4) mm Hg.”

The cytokine level changes demonstrate heterogeneous trend in different publications. How do authors account for it? How do the authors explain discrepancy?

Thank you for this interesting comment. This type of heterogeneity e.g. introduced by the study of Telini et al in patients in hemodialysis has been our concern as well. First, we tried to contact authors in order to verify the data. Second, when possible, we performed sensitivity analysis by excluding these studies (as discussed above in the preceding Q &A – modifications in the revised text have been, in line 383: “First, the available studies have been conducted in populations with different characteristics and types of chronic diseases. The existence of chronic diseases might have also influenced participants’ level of inflammation.). Given the fact that several diseases modify the immune and inflammatory response we believe that it is crucial that future studies will address this question by comparing the effect in healthy as well as in various diseased populations.

What do the rodents study confer? Does those results is in line with any of the published literature that has been discussed?

In the present review we deal only with human data. However, as the reviewer correctly questions the data on the same topic from animal models have been used in our introduction (Line 55: “Since then, several studies have been conducted in humans but mostly in animal models to attest the hypothesis of sodium induced systemic inflammation [4]. In animal models, the majority of the available data, but not all [5, 6] indicate that high sodium intake is associated with increased levels of circulating inflammatory biomarkers [7-10].”) to reinforce the hypothesis of sodium induced inflammation. However, by no means these examples should be interpreted and of course in paper we do not argue definitely in favor of translating these animal data to humans. Such data are always used as hypothesis generating and should be tested in human studies.

Please write few lines on the prospective experiments that would address the current issue. How would the trial look like? How does the meta-analysis help in designing new experiments that would answer the question?

We think that this is one of the merits of the present analysis since it will allow us and other researchers to identify methodological problems that can be addressed in future studies. 

The following text is included in the abstract (line 37): “Based on these identified methodological issues we propose that future RCTs should focus on young healthy participants to avoid confounding effects of comorbidities, should have 3 instead of 2 arms (very low, “normal” and high) daily sodium intake with more than 100 participants per arm, whereas an intervention duration of 14 days is adequate.

Also, in the discussion, (line 395) the following text is included: “Based on previously discussed data, we recommend that such RCT should have 3 arms (very low, “normal” and high) daily sodium intake, with sample size more than 100 participants per arm, focus on healthy and young participants to avoid the confounding effect of aging and comorbidities. A short study duration of 14 days with 7 days run-in period has been proved sufficient, and seems to be more adequate than longer studies, in order to accomplish maximal adherence to the study diets which however should be very closely monitored and verified with 24 repeated 24-hour urine selections. Most importantly, the above studies were all focused on circulatory biomarkers of inflammation and cannot provide evidence on tissue response or other types of inflammatory (e.g. cellular) response. Therefore, future studies should not only use state-of-the-art methodology to assess circulating biomarkers of inflammation but also to provide evidence at cellular level.

This manuscript is a resubmission of an earlier submission. The following is a list of the peer review reports and author responses from that submission.

Round 1

Reviewer 1 Report

Basdeki et al perform a review  on sodium intake an inflammation. There were only 12 studies eligible for inclusion. Even tough the the topic is of great interest big studies are pending. There were only 12 studies eligible for inclusion. Unfortunately, a metaanalysis including only 12 studies has no high impact.

Reviewer 2 Report

The manuscript entitled “Does sodium intake induce systemic inflammatory response? Α 2 systemic review and meta-analysis of randomized studies in 3 humans” by Basdeki et al., aims to show association between high salt intake and low salt intake with inflammatory biomarkers and blood pression.

 Interestingly, this systematic meta-analysis analyzes different manuscript looking at the inflammatory biomarkers as CRP, pro inflammatory cytokines (IL-1, IL-6, INF-g, TNF and IL-8) in salt intake dependent concentration.

The subject of this meta-analysis is highly critical in terms of the results because would confirm that excess in salt intake is harmful to health since associated with inflammation.

Although, the aim of this project should bring to a conclusion either positive either negative association it falls in between meaning does not add any interesting piece to the story. I fully understand that is difficult to dig inside the RCT and recognize which add to the study but if the authors did not find them suitable better to stop. I believe that everything is knowledge but , in my personal opinion, the methodology and the results of this paper do not fulfill criteria for publication

Reviewer 3 Report

Dear authors,

I found the study topic really interesting.

The abstract is quite clear and shows relevant information for the readers and the conclusiones are honest based on the results.

The introduction used good evidence from both sides about sodium intake. I miss one sentence or more talking about the differences between sodium included in high processed food VS non processed food in the introduction and its possible impacts on health.

The aim is clear and the study type is correct to solve it. 

The methods described follows PRISMA and show all the relevant information. 

In table 2 in the results you indicate that the duration of one study is 0.1 days. What is the meaning of that? Please specify the duration in hours if it is less than a day.

In Figure 2 it should be important two show the correlation between the points and the associated study.

In the forestplots it is not clear in which size is located HSD o LSD.

The limitations of the study must be included in the discussion.

Kind regards